# Neural Colour Correction for Indoor 3D Reconstruction Using RGB-D Data

**DOI:** 10.3390/s24134141

**Published:** 2024-06-26

**Authors:** Tiago Madeira, Miguel Oliveira, Paulo Dias

**Affiliations:** 1Institute of Electronics and Informatics Engineering of Aveiro (IEETA), Intelligent System Associate Laboratory (LASI), University of Aveiro, 3810-193 Aveiro, Portugal; tiagomadeira@ua.pt (T.M.); mriem@ua.pt (M.O.); 2Department of Electronics, Telecommunications and Informatics (DETI), University of Aveiro, 3810-193 Aveiro, Portugal; 3Department of Mechanical Engineering (DEM), University of Aveiro, 3810-193 Aveiro, Portugal

**Keywords:** neural network, colour correction, 3D reconstruction

## Abstract

With the rise in popularity of different human-centred applications using 3D reconstruction data, the problem of generating photo-realistic models has become an important task. In a multiview acquisition system, particularly for large indoor scenes, the acquisition conditions will differ along the environment, causing colour differences between captures and unappealing visual artefacts in the produced models. We propose a novel neural-based approach to colour correction for indoor 3D reconstruction. It is a lightweight and efficient approach that can be used to harmonize colour from sparse captures over complex indoor scenes. Our approach uses a fully connected deep neural network to learn an implicit representation of the colour in 3D space, while capturing camera-dependent effects. We then leverage this continuous function as reference data to estimate the required transformations to regenerate pixels in each capture. Experiments to evaluate the proposed method on several scenes of the MP3D dataset show that it outperforms other relevant state-of-the-art approaches.

## 1. Introduction

Three-dimensional (3D) reconstruction is the creation of 3D models from the captured shape and appearance of real objects. It is a field that has its roots in several areas within computer vision and graphics, and has gained high importance in others, such as architecture, robotics, autonomous driving, medicine, agriculture, and archaeology. With the rise in popularity of different human-centred applications requiring photo-realistic 3D reconstruction, such as virtual reality (VR) and augmented reality (AR) experiences, the problem of generating high-fidelity models has become an important task.

In a multiview acquisition system, particularly for large indoor scenes, the acquisition conditions, such as lighting, exposure, white balance, and other camera parameters, will differ along the environment, causing colour differences between captures and unappealing visual artefacts in the produced models. Different colour correction methodologies have been proposed for image stitching [1,2,3] algorithms. The global optimization strategy is widely used, and the colour correspondences are typically calculated based on the overlapping area between the images. When it comes to 3D reconstruction, especially through sparse RGB-D captures, this type of methodology cannot be directly applied, as the pose of the cameras will vary significantly in 6DOF within the environment. In this case, dense matching can be employed [4], but it is a very resource intensive process.

We propose a neural-based approach, using a multilayer perceptron (MLP) to learn an implicit representation of the colour in 3D space, while capturing camera-dependent effects. Then, to estimate the transformations required to regenerate pixels in each capture, one smaller MLP is trained using information provided by the larger MLP as reference colour. In the context of 3D reconstruction, even with a reduced number of pixel correspondences between images, a global optimization has scalability issues, as the combinations of images are exponential and processing of the captures required for an accurate reconstruction of an indoor scene can quickly become unfeasible. Conversely, our approach uses lightweight representations and is efficient in the integration process, using surface information derived from RGB-D data to create the continuous function of colour. Experiments to evaluate the proposed method on several scenes of the MP3D dataset show that it outperforms other relevant state-of-the-art approaches.

The remainder of this document is organized as follows: in Section 2, the related work is presented; in Section 3 the proposed methodology is described; in Section 4, the results are showcased and discussed; finally, Section 5 provides concluding remarks.

## 2. Related Work

In this section, the topic of colour correction is introduced, starting with pairwise techniques, also known as colour transfer, then going over colour consistency correction for image sets.

### 2.1. Pairwise Colour Correction

Depending on their foundational mapping principle, methodologies for colour correction of image pairs, or colour transfer, can be systematically categorized into two classes: nonparametric and parametric.

The majority of nonparametric methods are based on the direct mapping of all colour levels using a 2D joint image histogram (JIH). This is calculated from the known colour correspondences between the two images, often relying on overlapped regions. Pitié et al. [5] proposed a nonlinear iterative method, based on the one-dimensional probability density function transfer, mapping one N-dimensional distribution to another to create a colour mapping table. In [6], Su et al. used a gradient-aware decomposition model to separate the source image into several detail layers and base layers, performing colour transfer for each base layer using the previous methodology [5]. The base layers were then merged with boosted detail layers to generate the final output. De Marchi et al. [7] utilized a Vandermonde matrix to model the colour mapping function as a polynomial. Hwang et al. [8] combined moving least squares with the probabilistic modelling of the colour transfer in 3D colour space. Liu et al. [9] used unsupervised deep learning to separate the image into reflectance and shading components. Palette-based recolouring was then applied to the reflectance element, preventing colour overflow. Because the previously mentioned approaches are unable to fix local colour difference, local colour correction methods have been proposed. Wu et al. [10] used a content-based local colour transfer algorithm, integrating the spatial distribution of target colour style in the optimization process. Finlayson et al. [11] introduced a root-polynomial regression, invariant to scene irradiance and to camera exposure. Hwang et al. [12] improved upon their previous algorithm [8] by proposing bilateral weights, performing local colour transfer using spatial variation of colours. Niu et al. [13] perform a three-step correction: a coarse-grain colour correction for global colour matching, a fine-grain colour correction to improve both global and local colour consistency, and a guided filtering process to ensure structural consistency.

Unlike nonparametric methods, parametric methods describe the relationship between the colours of two images as a global transformation. The seminal work of Reinhard et al. [14] presented a global colour transfer using the mean and standard deviation. They employed the 
lαβ
 colour space in an attempt to untangle the different colour channels. Xiao and Ma [15] attempted to preserve the image gradient by leveraging the histogram of the target image and the gradient map of the source image. Nguyen et al. [16] performed a white-balance step on both source and target images to remove colour casts caused by different illuminations. They then corrected the global brightness difference by applying Xiao and Ma’s method [15] along the aligned white axis. He et al. [17] used semantic correspondences between the source and target images, leveraging neural representations and iteratively optimizing a local linear model to satisfy both local and global constraints. Wu et al. [18] segmented the images into regions of salient and nonsalient pixels using a weighted attention map. They then employed Reinhard et al.’s method [14] to correct these regions in the three channels of the YUV colour space. Lee at al. [19] proposed a deep neural network that leverages colour histogram analogy for colour transfer, modulating the colour of the source image with abstract colour features of the reference image. Even though there is no straightforward approach to the application of pairwise colour correction methods to the problem of harmonizing the colour of multiple images, it is evident that they have facilitated the development of colour consistency correction for image sets [20].

### 2.2. Colour Consistency Correction

Colour consistency correction is fundamental in multiview imaging systems and for applications such as 3D reconstruction, image mosaicking, or panoramic stitching. There are two predominant kinds of methodologies: those based on path propagation and those employing a global optimization. Path propagation approaches entail the selection of a reference image, and the progressive transfer of colour information to other images through a selected propagation path. Pan et al. [21] determined this path based on the dimension of the image overlap and time of transfer, while employing a global to local policy to improve the efficacy of the algorithm. Chen et al. [22] defined both the propagation path and reference image by minimizing the normalized error using the shortest distance algorithm. Dal’Col et al. [23] used the reprojection of geometric data to calculate accurate mappings between images and applied this information both on the computation of the shortest propagation path and in the calculation of the JIH colour mapping function. In these approaches, cumulative errors in the colour propagation can cause significant degradation of the results, particularly if there are pairs of images with low or incorrect mappings between them. The lack of a clear standard for the selection of a reference image is also a reasoned downside.

Global optimization methods solve for the parameters of all images simultaneously, minimizing the colour difference in corresponding regions. Each image is assigned an independent colour mapping model, and the colour consistency correction problem is formalized as a global energy optimization problem. The optimal solution of the energy function provides the final parameters of the image models, which correct the original images. Brown and Lowe [1] used a global gain compensation to minimize the colour difference in the overlapped regions. This approach was proposed for image stitching and has been implemented in the OpenCV stitching model. Xiong and Pulli [24] performed colour correction in the YCbCr colour space, applying gamma correction for the luminance component and linear correction for the chrominance components. Xia et al. [2] parameterized the colour remapping curve as a transform model, expressing the constraints of colour consistency, contrast and gradient in a uniform energy function. Yu et al. [25] combined global and local optimization strategies. They formalized the colour difference elimination as a least squares optimization problem, and mitigated local colour differences in the overlap areas of the target images with the gamma transform algorithm. Xie et al. [26] used a path propagation method to obtain a first guess for their global optimization approach. Keeping the distance between the corrected colour and the initial solution as a constraint on the optimization process, they attempted to avoid solutions that drift to unnatural appearance in the images. HaCohen et al. [4] used dense matching between images as colour correspondences to apply a global energy cost function. Their work focused on interactive colour correction of photo collections by automatically propagating changes made by a user to other images. Moulon et al. [27] employed the virtual line descriptor filter to augment the colour correspondences produced using feature matching and estimate global transformations for each colour channel independently, minimizing the colour difference through histogram matching. This method has been applied in OpenMVG. Shen et al. [28] proposed to accurately calculate overlapping regions by reprojecting a generated 3D mesh to the images. Park et al. [29] presented a robust low-rank matrix factorization method to estimate white balance and gamma correction parameters for each input image. Yang et al. [30] employed a spline curve remapping function, minimizing the variance in the colour values of each observation of the sparse points generated by structure from motion. Li et al. [20] proposed to group images using a graph partition algorithm. They performed intragroup correction and intergroup correction in sequence in an attempt to maintain accuracy and efficiency in the colour correction of large-scale images.

## 3. Proposed Approach

We propose a novel approach to tackle the problem of colour consistency correction, not by calculating mapping functions between pairs of images, nor by global optimization of camera models. Inspired by recent advances in neural implicit functions, such as DeepSDF [31], Fourier features [32], and NeRF [33], we represent a continuous scene as a 6D vector-valued function whose input is a 3D point location 
$\left(x_{p},y_{p},z_{p}\right)$
 and a 3D camera position 
$\left(x_{c},y_{c},z_{c}\right)$
, both under world coordinates, and whose output is a colour 
(r,g,b)
. We approximate this continuous 6D scene representation with an MLP network,

(1)
$$E_{\Theta }:\left(x_{p},y_{p},z_{p},x_{c},y_{c},z_{c}\right)\to \left(r,g,b\right)$$

and optimize its weights (
Θ
) to map from each input 6D coordinate to its corresponding colour, see Figure 1. The loss is a simple MSE error between each pixel in each image and the colour predicted by the network for the corresponding 3D point as seen by the specific camera,

(2)
$$L=\sum_{c\in C}\sum_{p\in P_{c}}{∥\hat{C}\left(p,c\right)-C\left(p,c\right)∥}_{2}^{2}$$

where 
C
 is the set of cameras of the dataset, 
$P_{c}$
 is the subset of 3D points that are seen by camera 
c
, 
$\hat{C}\left(p,c\right)$
 is the estimated colour, and 
C(p,c)
 is the reference colour for the pixel corresponding to the projection of point 
p
 in camera 
c
.

However, as discussed in [32], deep networks are biased towards learning lower-frequency functions, making it impossible to capture the high-frequency detail of the textures in 3D space. Therefore, we map our input to a higher dimensional space using the positional encoding:
(3)
$$\gamma \left(v\right)=\left(v,\sin\left(2^{0}\pi v\right),\cos\left(2^{0}\pi v\right),\cdots ,\sin\left(2^{L-1}\pi v\right),\cos\left(2^{L-1}\pi v\right)\right)$$

where *v* corresponds to each individual component of input 
$\left(x_{p},y_{p},z_{p},x_{c},y_{c},z_{c}\right)$
. The maximum frequency, *L*, was set to 10 for the point location, 
$\left(x_{p},y_{p},z_{p}\right)$
, and 4 for the camera location, 
$\left(x_{c},y_{c},z_{c}\right)$
.

We follow a similar architecture to DeepSDF [31] and NeRF [33], including a skip connection that concatenates the 3D point position to the fifth layer’s activation, as this seems to improve learning; see Figure 2. Additionally, by concatenating information on camera position only towards the end of the network, we encourage interpolation of the colour between cameras, capture shared colour, and complement information on 3D space that is not seen from that camera more effectively. Unlike radiance field models, which rely on consistent colour information across views to synthesize realistic colour, we use the camera information to capture differences between acquisition conditions, so that we may generate colour for any 3D point as if it were seen by a particular camera. In those approaches, volumetric rendering has become common, but since our goal is efficient colour harmonization for images used in texture mapping of 3D mesh models, we choose instead to use the geometric information provided by RGB-D captures, mapping colour directly to the surface coordinates. This way we avoid ray-sampling and all the computation required to find empty space, which for indoor reconstruction is most of the scene.

For each capture, we estimate a colour mapping function that will transform the RGB from the image to a set of colours consistent with the implicit representation of the colour in 3D space. This is achieved by training one smaller MLP per image, which will approximate the transformation required to regenerate the pixels for a particular capture,

(4)
$$D_{\Theta }:\left(r,g,b,x,y\right)\to \left(\hat{r},\hat{g},\hat{b}\right)$$

where 
x,y
 correspond to the col and row of the pixel with colour 
r,g,b
 in any given capture, and 
$\hat{r},\hat{g},\hat{b}$
 represent the estimated values for corrected colour, see Figure 3. Since these networks are mapping from one texture to another, they do not require positional encoding to elevate input to a higher dimensionality, see Figure 4. This approach makes our method robust to decimation of the depth data used to train the implicit scene representation. When regenerating pixels, these smaller MLPs will recover texture detail that may have been lost in the continuous scene representation, requiring only enough information to estimate a colour transfer from each capture to a corrected version that matches the colour learned by the MLP in the sampled points. The loss function for one MLP, learning the colour transformation for a capture, is

(5)
$$L=\sum_{i\in I}{∥\hat{C}\left(i\right)-C\left(p,c\right)∥}_{2}^{2}$$

where 
I
 is the set of pixels in a capture, 
$\hat{C}\left(i\right)$
 is the estimated colour for pixel 
i
, and 
C(p,c)
 is the reference colour for the pixel, obtained from the continuous scene representation 
$E_{\Theta }$
, by providing the corresponding 3D point 
p
 and a reference camera 
c
.

## 4. Results

In order to quantitatively assess colour consistency within the image sets, we adopt the commonly used PSNR and CIEDE2000 metrics. We calculate the colour similarity between images using the pairs of corresponding pixels, which we compute robustly using geometric data. Since both PSNR and CIEDE2000 are pairwise image metrics, we carry out the evaluation over each image pair in the set. The displayed score values include the mean and standard deviation for all image pairs. Additionally, we present a weighted mean based on the number of pixel correspondences.

CIEDE2000 [34] was adopted as the most recent colour difference metric from the International Commission on Illumination (CIE). This metric measures colour dissimilarity, meaning that lower score values equate to more similar colour measurements. The CIEDE2000 metric can be formulated as follows:
(6)
$$\mathrm{CIEDE}2000=\sqrt{\Delta L^{2}+\Delta C_{ab}^{2}+\Delta H_{ab}^{2}+R_{T}\cdot \Delta C_{ab}\cdot \Delta H_{ab}}$$

where 
ΔL
, 
$\Delta C_{ab}$
, and 
$\Delta H_{ab}$
 are the lightness, chroma, and hue differences, respectively, and 
$R_{T}$
 is a hue rotation term.

PSNR [35] measures colour similarity, meaning higher score values equate to more similar colour measurements. The PSNR formula is given by

(7)
$$\mathrm{PSNR}=10\cdot \log_{10}\left(\frac{L^{2}}{\mathrm{MSE}}\right)$$

where *L* is the largest possible value in the dynamic range of an image, and MSE is the mean squared error between the two images.

We showcase the results of our approach in three distinct scenes from the publicly available Matterport3D (MP3D) [36] dataset. We perform comparisons with several well-known state-of-the-art approaches, providing them with accurate and robust mappings between pixels, calculated by the reprojection of geometric data to each image. For [19], no pixel mappings are required, as colour histogram analogy is employed. In the case of pairwise approaches [7,11,14], we compute a graph of the image set and weigh the edges using the number of pixels mappings between captures. The reference image is decided by selecting the node with the largest centrality. We then compute the propagation paths by applying the shortest path algorithm, as suggested in [22,23].

Because of the nature of 3D reconstruction datasets, particularly those employing stationary sensors, captures can be sparse, with some image pairs presenting insufficient pixel correspondences to accurately calculate colour transfer. This can deteriorate results and cause excessive errors, particularly when applied through propagation paths. Furthermore, for optimization approaches [1,27], using every possible pair of images, regardless of the usefulness of their information, can quickly become unfeasible due to the exponential nature of combinations. Experimentally, we found that setting a small threshold relative to the resolution of the geometric data to avoid using image pairs with negligible information was beneficial for both the efficacy and efficiency of the tested colour correction approaches. For the results presented in this section, we discarded image pairs connected by less than 1% of the available geometric data.

Pairwise methods [7,11,14] have problems with accumulated error, despite the filtration process that was applied. Optimization-based approaches [1,27] seem to have trouble balancing a large number of images, particularly from such different perspectives and with large colour differences. The local colour discrepancies also make it impossible for the camera models to correctly compensate for the effects in the captures. Results for the colour balancing using histogram analogy [19] are somewhat underwhelming; this may be attributed to the fact that this approach is intended for more significant colour transfer and is suitable for problems involving style transfer between pairs of images rather than precise harmonization in a set. Our approach, leveraging implicit representation that takes into account the 3D space in which the textures will be applied, achieved the best results in all experiments; see Table 1, Table 2 and Table 3. As shown in Figure 5, Figure 6 and Figure 7, the proposed approach effectively harmonizes textures from different captures, significantly mitigating the appearance of texture seams. It should be noted that, while pixel regeneration is highly effective in creating consistent textures across the environment, it can sometimes result in a loss of high-frequency detail in certain areas. Several factors influence the final output, including the quality of registration between captures, the precision of sampling during training, the maximum frequency of positional encoding, the number of epochs, and other hyperparameters. These factors can lead to variations in texture detail and colour accuracy. Depending on the conditions of the input data, there is an inherent trade-off between the harmonization of the captures and the level of high-frequency information for particular regions.

## 5. Conclusions

In this paper, we propose a novel neural-based method for colour consistency correction. We leverage an MLP to learn an implicit representation of the environment colour in 3D space, while capturing camera-dependent effects. We then use this network to train smaller MLPs, approximating the required transformations to regenerate pixels in the original captures, producing more consistent colour. Our method is efficient and offers better scalability than global optimization approaches, which are the current most popular approach. Experiments show that the proposed solution outperforms other relevant state-of-the-art methods for colour correction.

The algorithm was designed within the scope of a 3D reconstruction pipeline, as a way to increase the quality of the generated models, making use of the available registered scanning information. However, this can limit its use in other applications. Also, since the proposed approach thrives on thorough depth information, poor-quality data may undermine the effectiveness of the algorithm. Nevertheless, as depth sensors improve and become more widespread, we can expect our method’s performance to enhance.

As future work, it would be interesting to explore anomaly detection techniques to identify points in 3D space with high variability in view-dependent colour. This approach could help pinpoint surfaces likely to exhibit more reflective properties. This information could be used to render the materials more accurately through physically-based rendering (PBR) and further enhance the colour correction results.

## Figures and Tables

**Figure 1 sensors-24-04141-f001:**
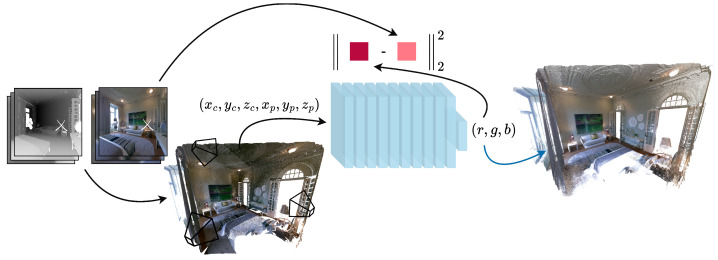
Overview of the training of our continuous scene representation using an MLP to approximate the colour in 3D space. Blue arrow represents a discrete sampling of the implicit function.

**Figure 2 sensors-24-04141-f002:**
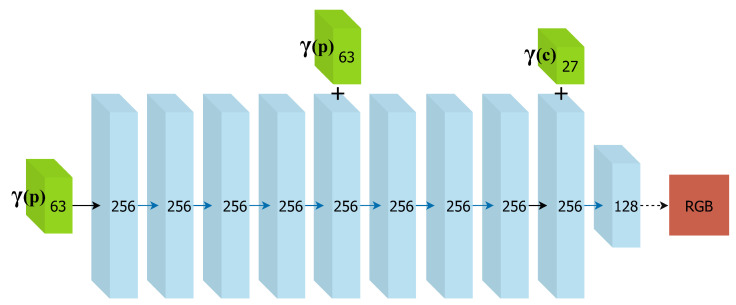
Visualization of our neural network architecture for implicit scene representation. Input vectors are shown in green. The number inside each block signifies the vector’s dimension. All layers are standard fully-connected layers, blue arrows indicate layers with ReLU activations, dashed black arrows indicate layers with sigmoid activation, and “+” denotes vector concatenation.

**Figure 3 sensors-24-04141-f003:**
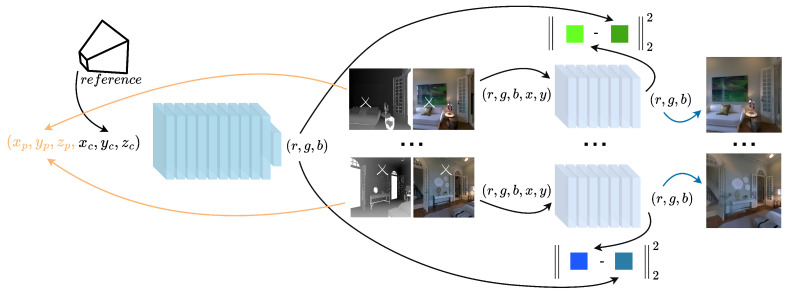
Overview of the training of our colour mapping function approximators, using the continuous scene representation as reference. The MLP previously trained to estimate colour in 3D space is used to provide ground truth for each of the smaller MLPs that will hold the approximation to regenerate pixels in the captures. Blue arrow represents the final output of the system.

**Figure 4 sensors-24-04141-f004:**
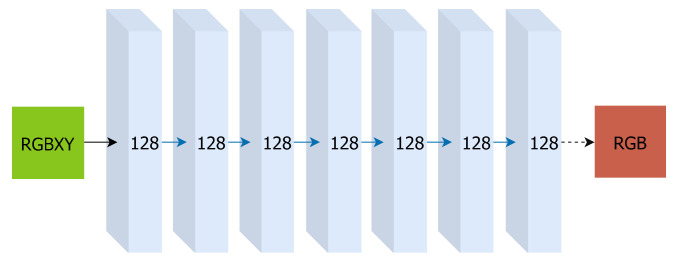
Visualization of our neural network architecture for individual capture colour transformation. Input vectors are shown in green. The number inside each block signifies the vector’s dimension. All layers are standard fully-connected layers, blue arrows indicate layers with ReLU activations, dashed black arrows indicate layers with sigmoid activation.

**Figure 5 sensors-24-04141-f005:**
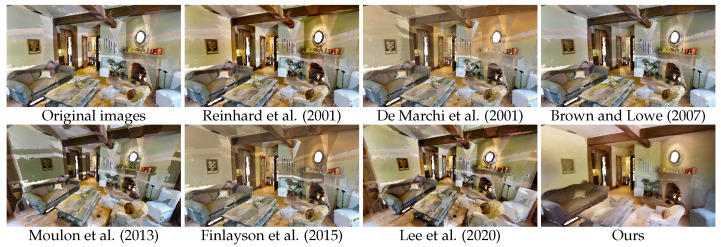
Qualitative comparison of our method with relevant state-of-the-art in scene 2azQ1b91cZZ_7 of the MP3D dataset. We observe that our method presents a more visually appealing mesh, with significantly reduced texture seams [1,7,11,14,19,27].

**Figure 6 sensors-24-04141-f006:**
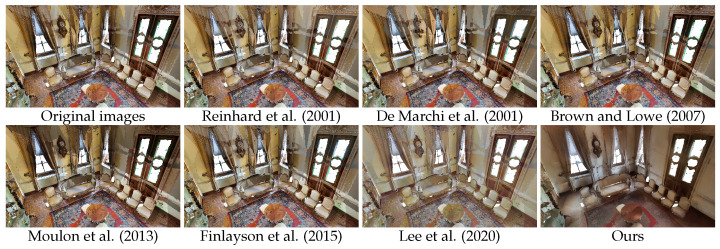
Qualitative comparison of our method with relevant state-of-the-art in scene VLzqgDo317F_19 of the MP3D dataset. We observe that our method presents a more visually appealing mesh, with significantly reduced texture seams [1,7,11,14,19,27].

**Figure 7 sensors-24-04141-f007:**
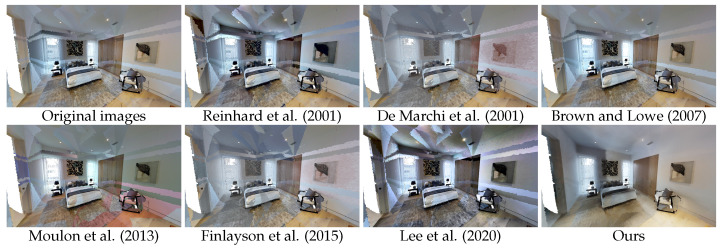
Qualitative comparison of our method with relevant state-of-the-art in scene aayBHfsNo7d_1 of the MP3D dataset. We observe that our method presents a more visually appealing mesh, with significantly reduced texture seams [1,7,11,14,19,27].

**Table 1 sensors-24-04141-t001:** Evaluation of colour consistency correction in scene 2azQ1b91cZZ_7 of the MP3D dataset. We observe that our method improves in all metrics, highlighted in bold.

	CIEDE2000 ↓			PSNR ↑		
Method	μ	$\mu_{w}$	σ	μ	$\mu_{w}$	σ
Original images	10.87	10.71	2.66	18.44	18.36	2.51
Reinhard et al. [14]	10.52	10.33	2.39	18.30	18.29	2.34
De Marchi et al. [7]	8.96	9.14	4.05	20.20	19.92	4.27
Brown and Lowe [1]	10.38	10.15	2.24	19.10	19.07	2.15
Moulon et al. [27]	6.49	6.41	1.37	23.73	23.71	2.19
Finlayson et al. [11]	8.77	8.84	2.93	20.12	19.96	3.00
Lee et al. [19]	11.39	11.17	2.66	17.94	17.99	2.30
Ours	**3.36**	**3.45**	0.78	**26.77**	**26.54**	2.66

**Table 2 sensors-24-04141-t002:** Evaluation of colour consistency correction in scene VLzqgDo317F_19 of the MP3D dataset. We observe that our method improves in all metrics, highlighted in bold.

	CIEDE2000 ↓			PSNR ↑		
Method	μ	$\mu_{w}$	σ	μ	$\mu_{w}$	σ
Original images	15.20	15.19	4.85	15.55	15.56	2.29
Reinhard et al. [14]	10.99	11.00	2.37	17.87	17.85	1.81
De Marchi et al. [7]	12.41	12.44	4.79	17.53	17.51	3.33
Brown and Lowe [1]	13.51	13.53	3.97	16.71	16.73	1.89
Moulon et al. [27]	7.24	7.22	1.52	21.50	21.55	1.80
Finlayson et al. [11]	11.19	11.22	3.46	17.81	17.77	2.36
Lee et al. [19]	12.11	12.10	2.09	17.38	17.38	1.34
Ours	**3.59**	**3.57**	1.23	**26.83**	**26.86**	2.79

**Table 3 sensors-24-04141-t003:** Evaluation of colour consistency correction in scene aayBHfsNo7d_1 of the MP3D dataset. We observe that our method improves in all metrics, highlighted in bold.

	CIEDE2000 ↓			PSNR ↑		
Method	μ	$\mu_{w}$	σ	μ	$\mu_{w}$	σ
Original images	9.74	9.56	4.36	20.74	20.62	4.85
Reinhard et al. [14]	11.07	10.87	3.30	18.00	17.99	3.16
De Marchi et al. [7]	8.43	8.51	5.58	22.06	21.58	5.51
Brown and Lowe [1]	8.21	8.36	3.30	22.70	22.21	5.16
Moulon et al. [27]	7.32	7.74	3.64	24.57	23.82	6.34
Finlayson et al. [11]	7.68	7.79	5.48	22.86	22.31	5.96
Lee et al. [19]	13.00	12.67	3.38	16.44	16.56	2.33
Ours	**2.49**	**2.55**	1.26	**30.60**	**30.04**	6.38

## Data Availability

Data are contained within the article.

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
