# Peer review of "Neural Colour Correction for Indoor 3D Reconstruction Using RGB-D Data"

_sensors, 2024, doi:10.3390/s24134141_

Round 1

Reviewer 1 Report

Comments and Suggestions for Authors

In this paper, they propose a neural-based approach to color correction for indoor 3D reconstruction. 

1) In equation (1), are the 3D point coordinates under a global or local coordinate system?

2) In Figure 3, how to use equation (3) to generate a 63 or 27 dimension vector? I think it should be an even dimensional vector.

3) In Figure 4, input vectors are not shown in green.

4) In Figures 5 and 6, detailed textures are blurred or lost, such as marble table tops and carpet patterns. These can cause severe color distortion.

Author Response

We would like to thank the reviewer for their thorough analysis of our manuscript. We have carefully considered each of their comments and made the necessary revisions to address the concerns raised. Below, we provide a point-by-point response:

"1) In equation (1), are the 3D point coordinates under a global or local coordinate system?"

The 3D point coordinates are under a global coordinate system. We have added this clarification to the text in line 160. The revised sentence now reads:

“...whose input is a 3D point location (xp, yp, zp) and a 3D camera position (xc, yc, zc), both under world coordinates, and whose output is a colour (r, g, b).”

"2) In Figure 3, how to use equation (3) to generate a 63 or 27 dimension vector? I think it should be an even dimensional vector."

We appreciate the reviewer's request for further clarification on positional encoding. Positional encoding is a method used to transform input coordinates into higher-dimensional spaces, enabling the neural network to learn more complex functions. Specifically, we use Fourier features to generate these vectors. The odd dimensions arise from the specific implementation of these features, where the original input is concatenated with the generated features.

For point position, (xp, yp, zp) we get: 3 features + 3 features * 2 (cos and sin) * 10 (maximum frequency L) = 3 + 3 * 2 * 10 = 63.

For camera position, (xc, yc, zc) we get: 3 features + 3 features * 2 (cos and sin) * 4 (maximum frequency L) = 3 + 3 * 2 * 4 = 27.

We have added further clarification about the maximum frequency and have altered equation (3) to reflect the concatenation of the input v:

“γ(v) = ( v, sin ( 20πv ) , cos ( 20πv ) , ··· , sin ( 2L−1πv ) , cos ( 2L−1πv )) (3) 

where v corresponds to each individual component of input (xp, yp, zp, xc, yc, zc). The maximum frequency, L, was set to 10 for the point location, (xp, yp, zp), and 4 for the camera location, (xc, yc, zc).”

"3) In Figure 4, input vectors are not shown in green."

We thank the reviewer for their attention to detail. The figure has been updated accordingly.

"4) In Figures 5 and 6, detailed textures are blurred or lost, such as marble table tops and carpet patterns. These can cause severe color distortion."

We agree with the reviewer that in certain areas there is some loss of texture detail. We have found this to be related to the quality of the registration between captures, but can also be influenced by several factors - Fourier features help networks learn high frequency functions, but very detailed texture can still be hard to capture, in particular in cases where geometric registration is not very accurate. Parameters such as the maximum frequency of the positional encoding, the number of epochs, the precision of sampling for the training, and other hyperparameters may affect the final output. Adjustments in these parameters can lead to variations in texture detail and colour accuracy. Depending on the conditions of the input data, there is an inherent tradeoff between the harmonization of the captures and the level of detail of the texture in certain areas. While our method aims to balance these aspects, achieving optimal results across all scenarios remains challenging.

We have added this information into the discussion of results, as it now reads:

“... Our approach, leveraging implicit representation that takes into account the 3D space in which the textures will be applied, achieved the best results in all experiments, see Tables 1, 2, and 3. As shown in Figures 5, 6, and 7, the proposed approach effectively harmonizes textures from different captures, significantly mitigating the appearance of texture seams. It should be noted that, while pixel regeneration is highly effective in creating consistent textures across the environment, it can sometimes result in a loss of high-frequency detail in certain areas. Several factors influence the final output, including the quality of registration between captures, the precision of sampling during training, the maximum frequency of positional encoding, the number of epochs, and other hyperparameters. These factors can lead to variations in texture detail and colour accuracy. Depending on the conditions of the input data, there is an inherent tradeoff between the harmonization of the captures and the level of high-frequency information for particular regions.”

Reviewer 2 Report

Comments and Suggestions for Authors

Thank you for the manuscript and work you present. Some minor suggestions and notes are following.

Acronym MLP, firstly introduced in l:20 and L44 is not defined. Please add it, the same way you do for VR and Ar in l:32

Figure 3: please crarify that this MLP is the 'global' one

Gigure 4: please clarify that this MLP is for each camera / position

Fig 5,6,7: Some linear artefacts are observed in the images of the other methods, but not in yours. Can you please explain (in text rather in caption) why this is happening?

RGB-D usually refers to RGB and D refers to depth (distance) from camera projection center to the object, as represented by the pixel. Hence, it is a mere distance. In your implementation you are using (Xc, Yc, Zc, Xp, Yp, Zp), which of course relays the distance, but you also include direction, which is inherent to your approach. Therefore, you are using much more information rather simply the distance, which is implied by the RGB-D acronym and might be misleading for readers or future implementations of your method. Pllease consider altering the title to convey to the readers that you are using direction as well. 

Author Response

"Thank you for the manuscript and work you present. Some minor suggestions and notes are following."

We would like to thank the reviewer for their comments, which have been very helpful for improving the manuscript. Next, we reply to each of the points raised by the reviewer.

"1) Acronym MLP, firstly introduced in l:20 and L:44 is not defined. Please add it, the same way you do for VR and AR in l:32"

We thank the reviewer for paying close attention. We have chosen to avoid referring to the MultiLayer Perceptron (MLP) in the abstract, describing the network instead:

“Our approach uses a fully connected deep neural network to learn an implicit representation of the colour in 3D space...”

And then define MLP on first use (L:45):

“…using a MultiLayer Perceptron (MLP) to learn an implicit representation of the colour in 3D space…”

"2) Figure 3: please clarify that this MLP is the 'global' one. Figure 4: please clarify that this MLP is for each camera / position"

We have updated the respective captions:

“Visualization of our neural network architecture for implicit scene representation…”

“Visualization of our neural network architecture for individual capture colour transformation…”

"4) Fig 5,6,7: Some linear artefacts are observed in the images of the other methods, but not in yours. Can you please explain (in text rather than in caption) why this is happening?"

We agree with the reviewer that this information is important to include in the analysis of results. We have added the information in the corresponding section:

“Our approach, leveraging implicit representation that takes into account the 3D space in which the textures will be applied, achieved the best results in all experiments, see Tables 1, 2, and 3. As shown in Figures 5, 6, and 7, the proposed approach effectively harmonizes textures from different captures, significantly mitigating the appearance of texture seams.”

"5) RGB-D usually refers to RGB and D refers to depth (distance) from camera projection center to the object, as represented by the pixel. Hence, it is a mere distance. In your implementation you are using (Xc, Yc, Zc, Xp, Yp, Zp), which of course relays the distance, but you also include direction, which is inherent to your approach. Therefore, you are using much more information rather simply the distance, which is implied by the RGB-D acronym and might be misleading for readers or future implementations of your method. Please consider altering the title to convey to the readers that you are using direction as well."

We would like to thank the reviewer for being meticulous in their analysis of the scope of our method. In our manuscript, the title "Neural Colour Correction for Indoor 3D Reconstruction Using RGB-D Data" indicates the type of dataset we use as input. 

Radiance field models typically use only RGB data but rely on viewing direction, ray marching, and differentiable volumetric rendering to represent the scene. The input for training a Neural Radiance Field (NeRF) is position + direction (x,y,z,θ,ϕ) for each ray.

Conversely, our approach relies on geometric information from RGB-D data, which is what we intend to convey in the title. We do not utilize ray marching; instead, we train using only 3D point locations  (??, ??, ??) and 3D camera positions (??, ??, ??). Indeed, we can only obtain the point locations in a global reference frame from the depth data if the captures are correctly registered, which requires information about the pose of each camera. Despite this, we believe it is not misleading to describe a dataset of registered RGB-D captures as "RGB-D data" as this is commonplace. Most RGB-D datasets include camera pose, and sensor data capturing pipelines typically include algorithms to generate this information. Furthermore, since viewing direction is not used directly in training of our neural network, we believe it may be detrimental to add it to the title.

Round 2

Reviewer 1 Report

Comments and Suggestions for Authors

This article has been revised according to my comments. The article is clearer and easier to read.